# Assessment of the Influence of 5-Fluorouracil on SMAD4 and TGFB1 Gene Expression, Apoptosis Induction and DNA Damage in Human Cell Lines

**DOI:** 10.3390/bioengineering10050570

**Published:** 2023-05-09

**Authors:** Agnieszka Wosiak, Dagmara Szmajda-Krygier, Jacek Pietrzak, Joanna Boncela, Ewa Balcerczak

**Affiliations:** 1Laboratory of Molecular Diagnostics and Pharmacogenomics, Department of Pharmaceutical Biochemistry and Molecular Diagnostics, Medical University of Lodz, 1 Muszynskiego, 90-151 Lodz, Poland; 2Institute of Medical Biology, Polish Academy of Science, 106 Lodowa, 93-232 Lodz, Poland

**Keywords:** colorectal cancer, *SMAD4*, 5-fluorouracil, *TGFB1*, cell line, CACO-2, SW480, SW620

## Abstract

Purpose: Suppressor of mothers against decapentaplegic homolog 4 (SMAD family member 4, *SMAD4*) is involved in the adenoma–carcinoma pathway, leading to the development of colon cancer. The encoded protein is a key downstream signaling mediator in the TGFβ pathway. This pathway has tumor-suppressor functions, including cell-cycle arrest and apoptosis. Its activation in late-stage cancer can promote tumorigenesis, including metastasis and chemoresistance. Most colorectal cancer patients receive chemotherapy based on 5-FU as an adjuvant treatment. However, the success of therapy is hampered by multidrug resistance by neoplastic cells. In colorectal cancer, resistance to 5-FU-based therapy is influenced by *SMAD4* gene expression, as patients with decreased *SMAD4* gene expression probably have a higher risk of developing 5-FU-induced resistance. The mechanism leading to the development of this phenomenon is not fully understood. Therefore, the present study assesses the possible influence of 5-FU on changes in the expression of the *SMAD4* and *TGFB1* genes. Patients and methods: The effect of 5-FU on the expression of *SMAD4* and *TGFB1* in colorectal cancer cells derived from the CACO-2, SW480 and SW620 cell lines was evaluated using real-time PCR. The cytotoxicity of 5-FU on colon cancer cells was assessed by the MTT method, and its effect on the induction of cell apoptosis and the initiation of DNA damage using a flow cytometer. Results: Significant changes in the level of *SMAD4* and *TGFB1* gene expression were noted in the CACO-2, SW480 and SW620 cells treated with 5-FU at various concentrations during 24 h and 48 h exposure. The use of 5-FU at a concentration of 5 µmol/L resulted in a decrease in the expression of the *SMAD4* gene in all cell lines at both exposure times, while the concentration of 100 µmol/L increased the expression of the *SMAD4* gene in CACO-2 cells. The level of expression of the *TGFB1* gene was higher for all cells treated with 5-FU at the highest concentrations, while the exposure time was extended to 48 h. Conclusion: The observed in vitro changes in CACO-2 cells caused by 5-FU may be of clinical relevance when choosing the drug concentration for treating patients with colorectal cancer. It is possible that 5-FU has a stronger effect on colorectal cancer cells at the higher concentrations. Low concentrations of 5-FU may not have a therapeutic effect and may also influence drug resistance in cancer cells. Higher concentrations and prolonged exposure time may affect *SMAD4* gene expression, which may increase the effectiveness of therapy.

## 1. Introduction

Colorectal cancer (CRC) is one of the most common malignancies, being the third highest cancer-related cause of death globally. Despite continual research on the etiopathogenesis of CRC, a large percentage of patients diagnosed with colorectal cancer still die within five years following removal of the primary tumor due to cancer recurrence. Most cases of colorectal cancer are sporadic and are often diagnosed at an advanced stage of the disease. Understanding the molecular basis of colorectal cancer development has made it possible to use epigenetic and genetic changes as markers of cancer progression in clinical practice. This has contributed to the achievement of significant progress in the treatment of patients [1].

A common cytogenetic alteration observed in the development of colorectal cancer is the loss of genetic material in the region of the long arm of chromosome 18, where the *SMAD4*, *SMAD2* genes and other tumor suppressor genes are located. In addition, the presence of 18q21 chromosomal region deletion in colorectal cancer patients is associated with a worse prognosis and poor survival. This is probably related to a loss of the important function that the *SMAD4* gene fulfills in regulating the growth, proliferation, differentiation and apoptosis of epithelial cells [2]. The protein encoded by the *SMAD4* gene is a key downstream signaling mediator in the TGFβ pathway, which plays a complex role in the control of critical biological processes in the gastrointestinal epithelium. Disturbances in the TGFβ signaling pathway component can lead to the development and progression of tumors, including colorectal cancer. The TGFβ pathway can regulate the cell cycle in two ways, depending on the status of the SMAD and non-SMAD-related pathways [3]. The pathway mediators protect the epithelial cells from excessive growth by directing them to apoptosis and promoting cell differentiation; however, the same elements stimulate neoplastic cell proliferation [4,5].

The *SMAD4* gene, associated with the TGFβ pathway, is a tumor suppressor gene, and its loss disrupts TGFβ signaling. This gene plays a very important role in the progression of colorectal cancer, since it is responsible for regulating the transcription of many target genes, including inflammation-related genes and proapoptotic genes. Moreover, *SMAD4* modulates the expression of growth factors such as VEGF [6]. *SMAD4*-deficient colonic epithelial cells have the ability to proliferate and metastasize 1. The main cause of *SMAD4* loss in the CRC is loss of heterozygosity (LOH); however, in addition to its lack of expression, various other mechanisms also promote a more aggressive course of the disease [7,8]. *SMAD4* loss is also associated with chemoresistance to 5-fluorouracil (5-FU), used in the treatment of colorectal cancer. The correlation of *SMAD4* loss with a worse course of colorectal cancer development may be due to the promotion of chemoresistance to 5-FU in these patients [9,10,11].

Currently, the most effective therapeutic strategies for colorectal cancer include chemotherapy and radiation therapy following resection. Most treatments for cancer, despite beneficial effect, result in side effects which adversely affect patient quality of life (QOL), and it was found that clinical pharmacist intervention resulted in the reduction of treatment-related side effects and the improvement of patients’ QOL [12]. Additionally, a cross-sectional observational study performed on 500 cancer patients to identify the incidence of prescribing medication errors (PME) involving chemotherapeutic agents reported that all the cases contained at least one error, and the risk factors predicting the prescribing errors were the protocol type, the tumor type and the toxicity type of the antineoplastic regimen, which should be prevented for improvement of the treatment plan [13]. Chemotherapy based on 5-fluorouracil is used as an adjuvant treatment in most colorectal cancer patients who are at high risk of relapse [14,15]. A common and unresolved problem faced when implementing an adjuvant therapy in patients with colorectal cancer is the phenomenon of multidrug resistance. Fluorouracil is a pyrimidine analogue which demonstrates a cytotoxic effect after conversion into biologically active cell-cycle-specific metabolites. Three enzymes are involved in the metabolism of 5-FU in the body: dihydropyrimidine dehydrogenase, synthase thymidylate and methylenetetrahydrofolate reductase. It is believed that 5-FU exerts its toxic effect on cancer cells by damaging DNA and preventing nucleic acid synthesis through thymidylate synthase inhibition, resulting in cell-cycle arrest in the S phase, the inhibition of cell division and, ultimately, cell apoptosis [16].

The level of *SMAD4* gene expression plays a particular role in inducing resistance to 5-FU-based therapy. As such, the level of *SMAD4* mRNA in patients with advanced colorectal cancer could be used as a genetic marker of the response to adjuvant therapy with 5-FU. However, the mechanism by which *SMAD4* regulates the chemosensitivity of patients with CRC undergoing adjuvant treatment has not been fully investigated [9,17,18,19]. In addition, the level of gene expression may also depend on the concentration of the drug used [20].

Therefore, the aim of the present study was to evaluate the potential of 5-FU in treating late-stage disease by determining the effect of 5-FU on the viability of advanced colorectal cancer cell lines at different concentrations and whether 5-FU may influence *SMAD4* gene expression, which is a prognostic factor in patients with colorectal cancer, and it also plays an important role in the response to 5-FU treatment. Understanding the mechanism that regulates *SMAD4* gene expression can provide valuable information about the effectiveness of 5-FU in the therapy-selection process.

## 2. Materials and Methods

### 2.1. Material

The effect of 5-FU on the level of *SMAD4* and *TGFB1* gene expression and on apoptosis induction or DNA damage in colorectal cancer was determined for three human colon cancer cell lines: CACO-2, SW480 and SW620, purchased at Sigma-Aldrich (Hamburg, Germany). The company provides a certificate of authenticity for the cell lines (European Collection of Authenticated Cell Cultures). According to the manufacturer’s data, the CACO-2 cells were originally derived from colon tissue obtained from a 72-year-old Caucasian male diagnosed with colorectal adenocarcinoma. The selected CACO-2 cell line was characterized for different molecular markers; the cells were found to show no structural changes within the *SMAD4* gene. The SW480 cells were originally obtained from a 50-year-old man suffering from colorectal B adenocarcinoma, according to the Dukes classification; the cells were isolated from the primary tumor lesion and express epithelial growth factor. The SW620 cells are derived from the same patient as SW480, but were isolated from colorectal adenocarcinoma metastasis to the lymph node. They have a G > A mutation in codon 273 of p53, resulting in an amino acid substitution (His > Arg). The study plan is presented as additional material in Appendix A.

### 2.2. Methods

#### 2.2.1. Cell Culture

The CACO-2, SW480 and SW620 cell lines show adherent growth. The selected cell lines were cultured in flasks with Eagle’s Minimum Essential Medium (MEM) without phenol red, supplemented with 10% heat-inactivated fetal bovine serum (FBS) and gentamicin. All cell culture reagents were purchased from Genos (Lodz, Poland). The cells were grown in an incubator with a 5% CO_2_ atmosphere, at a temperature of 37 °C and appropriate air humidity. When cell confluence was 70–80%, the cell culture was passaged using trypsin. All 3 cell lines were analyzed in 3 independent replicates of experiments for each time point.

#### 2.2.2. Assessment of Cell Viability by MTT (3-(4,5-Dimethylthiazol-2-yl)-2,5-Diphenyltetrazolium Bromide) Assay

The MTT test was performed to evaluate the cytotoxicity effect of 5-FU on colorectal cells and to select an appropriate dose range of the studied drug. The assay was performed in a 96-well plate. Plates were seeded with a suspension of CACO-2, SW480 and SW620 cells at a density of 1.5 × 10^4^ cells/mL in a volume of 100 µL per well. After 24 h, the conditioned medium was removed from the plate and the test compound dissolved in fresh experimental culture medium was added to the wells in increasing concentrations. For this test, a 5-FU solution was used at the following concentrations: 1, 5, 10, 100, 1000 µmol/L [21]. Both plates were incubated for 24 h and 48 h of exposure to 5-FU. Following the end of incubation, the medium containing the test compound was removed and the cells were treated with a solution of MTT at a final concentration of 0.5 mg/mL. After a two-hour reaction at 37 °C, the formed formazan crystals were dissolved in a 20% solution of sodium lauryl dodecyl sulphate (SDS) in 0.01 M HCl. The plates were allowed to incubate. At the last stage, the plates were read colorimetrically by measuring the absorbance at a wavelength of 570 nm. The viability test was performed on 3 cell lines in 3 independent replicates for each time point. The results were used to confirm the effect of 5-FU on cell viability and select the concentrations of the test compound used in the next stages to determine the level of *SMAD4* and *TGFB1* gene expression after exposure to 5-FU.

#### 2.2.3. Assessment of Cell Death by MUSE Test Kit

The effect of 5-FU on the initiation of apoptosis in CACO-2 cells was assessed with a Guava Muse Cell Analyzer flow cytometer using the Muse^®^ Annexin V and Dead Cell Assay Kit (Luminex, Austin, TX, USA). This method allows the detection of four cell populations, viz. living cells, dead cells and early and late apoptotic cells, based on the binding of fluorescently labeled annexin V with phosphatidylserine (PS) located on the cell membrane. For the analysis, the cells of the CACO-2 line were cultured in six-well plates. After a 24 h incubation period, the cells were exposed to 0.1, 1, 10, 100 and 1000 µmol/L 5-FU solution for 24 h. After this time, the cells were harvested from the surface of the wells in the plate using trypsin. The obtained cell suspensions in the culture medium were used for cytometer assay according to the manufacturer’s instructions.

#### 2.2.4. Assessment of DNA Damage by MUSE Test Kit

The degree of DNA damage associated with the ATM-dependent pathway in the CACO-2 line under the influence of 5-FU was determined with a Guava Muse Cell Analyzer cytometer using a Flow Cellect Multi-Color DNA Damage Response reagent kit. The results allow the analyzed cells to be divided into four groups: cells without DNA damage, ATM (ataxia-telangiectasia-mutated)-activated cells, H2A.X-activated cells (cells with phosphorylation of the histone variant H2AX) and cells with double DNA strand breaks using fluorescent-based analysis. For the assay, CACO-2 cells were grown in six-well plates and treated with 5-FU at concentrations of 0.1, 1, 10, 100 and 1000 µmol/L. After a 24 h exposure, the cells were trypsinized from the plates into tubes and resuspended in culture medium, then centrifuged and washed with PBS (phosphate-buffered saline) buffer solution. The obtained cell suspensions in PBS were used for the assay according to the manufacturer’s instructions.

#### 2.2.5. Assessment of *SMAD4* and *TGFB1* Gene Expression in CACO-2, SW480 and SW620 Cells Treated with 5-FU

##### Exposure of Cells to 5-FU

The cells were exposed to 5-FU in two 6-well plates. First, a suspension of cells with a density of 1.5 × 10^4^ cells/mL was set up in the plates in a volume of 3 mL per well and left for 24 h. Following this, a series of dilutions of the test compound were prepared in antimicrobial-free culture medium. The prepared solutions of 5-FU were applied to the seeded wells at concentrations of 0.1, 1, 5, 10 and 100 µmol/L. Cells incubated with growth medium only and not treated with the test compound were used as negative control. After the set exposure time (24 h and 48 h), the old medium containing the test substance was removed from the plates and the cells treated with trypsin. Finally, the resulting test cell suspensions were taken from the wells into test tubes, centrifuged, washed twice with the buffer solution and, finally, suspended in PBS. The prepared cells treated with 5-FU were stored at −20 °C until RNA isolation was performed.

##### RNA Isolation from 5-FU-Treated Cells

In order to evaluate the effect of 5-FU on the expression of the SMAD4 and TGFB1 genes in the CACO-2, SW480 and SW620 cell lines, RNA was isolated using the column method with a Genomic Midi Kit (A&A Biotechnology, Gdańsk, Poland) in accordance with the manufacturer’s protocols. The concentration and purity of RNA obtained after isolation was assessed spectrophotometrically. RNA samples with a 260/280 nm absorbance ratio between 1.8 and 2.0 were selected for further study.

##### Reverse Transcription Reaction

Total RNA obtained from the 5-FU-treated cell lines was transcribed to cDNA using a High-Capacity cDNA Reverse Transcription Kit (Applied Biosystems™, Waltham, MA, USA) according to the manufacturer’s protocol. The final RNA concentration of 0.05 µg/µL was determined for each sample. The presence of cDNA in the samples was checked by PCR for the GAPDH reference gene; samples demonstrating the presence of the 189 bp PCR product were stored at −20 °C for further analysis.

##### Real-Time PCR

The levels of *SMAD4* and *TGFB1* mRNA were assessed by comparison to the GAPDH reference gene using the real-time PCR on a CFX Connect Real-Time PCR Detection System (Bio-Rad, Hercules, CA, USA) in accordance with the manufacturer’s protocol. The PCR reaction was performed with a nonspecific fluorescent dye, using the iTaq Universal SYBR Green Supermix reagent kit (Bio-Rad, Hercules, CA, USA). The composition of the reaction mixture for both genes comprised 5 μL of mix reagent, 0.25 μL of 10 µmol/L each primer solution, 1 μL of cDNA template and nuclease-free water to a final reaction volume of 10 μL. The sequences of the primers used for the reactions were GAPDH F 5′-TGGTATCGTGGAAGGACTCATGAC-3′; GAPDH R 5′-ATGCCAGTGAGCTTCCCGTTCAGC-3′; SMAD4 F 5′-GCCTGATCTTCACAAAAATG-3′; SMAD4 R 5′-GATCAATTCCAGGTGATACAAC-3′; TGFB1 F 5′-AACCCACAACGAAATCTATG-3′; TGFB1 R 5′-CTTTTAACTTGAGCCTCAGC-3′. Real-time PCR reactions for *SMAD4*, *TGFB1* and *GAPDH* were run simultaneously, in separate tubes and in triplicate, for each sample. A triplicate negative control amplification reaction was performed for each experiment. The thermal cycling conditions for the reaction comprised the following: initial denaturation step at 95 °C for three minutes, followed by 39 cycles with two steps, viz. 10 s denaturation at 95 °C and 30 s annealing with elongation at 58 °C. To confirm real-time reaction specificity, melting curve analysis was performed for each amplification product. The mean of the obtained Ct values for *GAPDH*, *TGFB1* and *SMAD4* genes was calculated. The reaction was carried out in three independent replicates. The relative level of *SMAD4* and *TGFB1* expression was assessed by the ΔΔCt method. The gene expression analysis was performed on 3 cell lines in 3 independent replicates for each time point.

#### 2.2.6. Bioinformatic Analysis of *SMAD4* Gene Expression

TCGA (The Cancer Genome Atlas) analysis data in the UALCAN database (http://ualcan.path.uab.edu/index.html, accessed on 12 January 2023) was used to compare the level of *SMAD4* gene expression between normal colon tissue and primary colon adenocarcinoma tissue. The results were compared using the Student’s t-test and presented as box plots. The RNA-seq results for the tissue types were also compared based on the Mann–Whitney U-test using the TNMplot tool (https://tnmplot.com, accessed on 12 January 2023), with the results given in the box plots.

The relationship between *SMAD4* gene expression and individual cancer stage in colorectal adenocarcinoma was determined using the UALCAN database. The statistical significance of any differences between the compared features was assessed with the Student’s t-test, with the results presented in box plots.

The study design does not have any features of a medical experiment or a clinical trial performed on a patient, and therefore is not subject to assessment by the Local Bioethics Committee of the Medical University of Lodz, Poland (decision no. RNN/190/22/KE of 12 July 2022). Hence, the present study, i.e., evaluation of *SMAD4* gene expression in colorectal cancer based on published results contained in public databases, can be carried out without the need for an opinion from the Bioethics Committee.

#### 2.2.7. Statistical Analysis

The obtained results were subjected to statistical analysis using STATISTICA 13.1. software (TIBCO, Palo Alto, CA, USA). The statistical significance was calculated using the ANOVA test with post hocTukey’s HSD test. The results of the relative expression of SMAD4 and TGFB1 are presented as mean +/−0.95 confidence interval. A *p*-value less than 0.05 was considered as statistically significant.

## 3. Results

### 3.1. The Results of the MTT Assay

The viability of the 5-FU-treated cells was assessed using the MTT test. The percentage viability was calculated as (drug-treated cells absorbance/control cells absorbance) ×100. The results indicate that the CACO-2, SW480 and SW620 cell lines showed resistance to 5-FU applied at 1–10 µmol/L, particularly during the 24 h exposure. Using these drug concentrations, the cells demonstrated similar viability to the control (untreated) cells. However, the highest 5-FU concentrations, i.e., 100 µmol/L and 1000 µmol/L 5-FU, had a significant effect on the viability of colorectal cancer cells (*p* < 0.005). In addition, extending the time of exposure to 48 h resulted in greater differences in the cell viability of all cell lines.

CACO-2 cells turned out to be the most sensitive to the action of 5-FU; concentrations of 1–1000 µmol/L resulted in a greater than 50% decrease in cell viability (*p* < 0.05). The SW620 cells demonstrated a significant decrease in viability below 65% for all 5-FU concentrations (1–1000 µmol/L) for 48 h. The SW480 cell line was found to be the most resistant to 5-FU treatment during the 48 h exposure: significant decreases in viability were only noted at concentrations of 10–1000 µmol/L. The results obtained in the test are presented in Figure 1 and Figure 2.

### 3.2. The Results of the Flow Cytometry Assays

The results of the flow cytometry tests are shown in Figure 3 and Figure 4. Based on the presented findings, it was confirmed that 5-FU does not significantly affect cell apoptosis or cause DNA damage when used in concentrations of 0.1–10 µmol/L. In CACO-2 cells, when 5-FU was applied at a concentration above 100 µmol/L, a higher percentage of cells were found to be dead or in the late stage of apoptosis: 12.45% of cells were dead at 100 µmol/L and 14.54% were dead at 1000 µmol/L.

Similarly, cells treated with >100 µmol/L showed more frequent changes related to DNA damage compared to control cells. At a concentration of 100 µmol/L, the number of cells with total DNA damage rose to 20.58%, compared to 5.71% for controls. At 1000 µmol/L, the percentage of dead cells increased to 29.5% compared to the control.

The percentage of cells with the single positive phosphorylated form of H2A.X (pH2A.X) was 9.23% and 19.4%, respectively, for 100 and 1000 µmol/L, compared to 2.6% in the control. The incidence of cells with double-strand breaks also increased to about 10% at the two highest concentrations of 5-FU.

### 3.3. The Relative Expression of the SMAD4 and TGFB1 Gene in Colorectal Cells Treated with 5-FU

In the quantitative analysis, the assumed level of reaction efficiency was 100%. The double-delta method was used to calculate the relative expression ratio (R) of the *SMAD4* and *TGFB1* genes. Significant changes in the level of *SMAD4* expression were noted in the CACO-2, SW480 and SW620 cells treated with 5-FU at various concentrations during 24 h and 48 h exposure compared to the untreated control cells (*p* < 0.05). A significant reduction in *SMAD4* gene expression was observed in cells of all lines following 24 h exposure with the 5-FU (5 µmol/L) compared to untreated control cells. Extending the exposure time to 48 h in the SW620 and CACO-2 lines produced a similar change in SMAD4 gene expression. In the case of CACO-2 cells, 48 h exposure to 5-FU reduced the SMAD4 gene expression compared to controls at concentrations ranging from 1–10 μmol/L, but increased it at 100 μmol/L. Similar observations were noted regarding the level of TGFB1 gene expression in CACO-2 cells after 24 h exposure: gene expression fell significantly for 1–100 μmol/L but significantly increased at 1000 µmol/L (all *p* < 0.05). During longer exposure, both 100 µmol/L and 1000 µm increased gene expression.

In the case of the remaining lines, SW480 and SW620, *TGFB1* gene expression was found to increase over the concentration range of 5–1000 µmol/L (48-h exposure), while only the concentration of 1 µmol/L appeared to reduce gene expression. No significant correlation between the treatment and the levels of *TGFB1* gene expression was observed following 24 h exposure in either SW480 or SW620 cells. The results are presented in Figure 5, Figure 6, Figure 7 and Figure 8.

### 3.4. Bioinformatics Analyses

SMAD4 gene expression was compared between normal and tumor colon tissue using the data contained in the UALCAN and TNMplot databases. Both analyses found SMAD4 gene expression to be significantly lower in colorectal cancer than in healthy intestinal tissue (*p* < 0.01)—Figure 9 and Figure 10. In addition, SMAD4 expression was significantly lower in stage 4 cancer compared to healthy colon tissue (*p* < 0.01)—Figure 11.

## 4. Discussion

Fluorouracil (5-fluorouracil) is widely used in the treatment of patients with colorectal cancer. A common problem limiting the application of a thymidylate synthase inhibitor in oncological therapy is chemoresistance, manifested by the lack of response of tumor cells to the chemotherapeutic agent. In colorectal cancer, the level of expression of the *SMAD4* gene plays a role in inducing resistance to 5-FU-based therapy; however, the molecular mechanism underlying this phenomenon remains unknown. Many scientific reports show that patients with reduced *SMAD4* gene expression are at higher risk of 5-FU-induced resistance. As high *SMAD4* expression is a desirable predictor in patients with colorectal cancer, especially in advanced stages, determining *SMAD4* expression before treatment can be used to estimate the potential benefits of the planned therapy.

In addition, the specific concentrations of 5-FU on *SMAD4* gene expression may also have a significant effect on therapy outcome. In a report on the role of the SMAD4 gene as a predictive marker of therapy based on 5-FU as adjuvant therapy, Boulay et al. indicate that patients with colorectal cancer and normal *SMAD4* expression gain significantly greater benefits [22]. The *SMAD4* gene plays a very important role in the development of colorectal cancer, and the loss of this gene increases the likelihood of progression to the metastatic form of this cancer. Pia Alhopuro et al. report a significant relationship between the low SMAD4 protein level in tumor tissue and decreased *SMAD4* mRNA expression, resulting in poor prognosis and a high risk of disease recurrence after treatment [11,23].

In addition, our search of the UALCAN and TNMplot databases found that a lower level of *SMAD4* gene expression correlates both with the development of colorectal cancer and with a higher cancer stage. Reduced *SMAD4* expression is associated with the activation of the PI3K/Akt pathway and increased VEGF expression in colon cancer cells, which promotes cancer progression and escape from apoptosis. Moreover, the level of *SMAD4* expression influences the overexpression of *BCL2* and *BCLW* genes responsible for inhibition of cell death, and, hence, the disruption of the apoptotic pathway [24,25]. It is believed that 5-FU acts by directing neoplastic cells into the apoptotic pathway by inhibiting thymidylate synthase. The correlation between 5-FU resistance and the level of *SMAD4* gene expression may suggest that this gene has a significant contribution the 5-FU-dependent apoptosis [26,27].

The present study evaluated changes in the expression of the *SMAD4* gene in advanced colorectal cancer cell lines under the influence of various concentrations of 5-FU. Our findings indicate that colorectal cancer cells were more sensitive to 5-FU after prolonged exposure to the drug. Additionally, significantly lower relative expression of the *SMAD4* gene was observed in CACO-2 cells treated for 48 h with 1–10 µmol/l 5-FU and for SW620 cells after 24 h exposure at the same concentrations compared to untreated cells. These observations may indicate the possibility of drug resistance in colorectal cancer cells treated with 5-FU at low concentrations.

Although the use of 5-FU at low concentrations may not yield a therapeutic effect, it may also have a negative result on tumor cells that become resistant. Moreover, it was shown that 48 h treatment with 100 µmol/L 5-FU resulted in a significant increase in the expression of the *SMAD4* gene in CACO-2 cells. This observation may indicate that 5-FU influences the expression of the SMAD4 gene, which may contribute to overcoming drug resistance in colorectal cancer cells. Based on these preliminary results, a more thorough analysis is needed to determine the effects on other colorectal cancer cell lines, and in in vivo models. Indeed, not all the concentrations of 5-FU used in the present study appeared to influence *SMAD4* gene expression, and in vivo studies may be better suited to determining an appropriate concentration of 5-FU in colorectal cancer cells for overcoming resistance, increasing the effectiveness of the treatment and, thus, significantly improving prognosis.

Changes in the *TGFβ* gene are considered an important contributor to the development of colorectal cancer. The TGF-β cytokine, in addition to its physiological effect on normal cells, has an antitumor effect in the early stages of carcinogenesis, inhibiting proliferation and enhancing cell differentiation. However, the dysregulation of the TGF-β signaling pathway supports the formation of metastases, neoangiogenesis and the phenomenon of the epithelial–mesenchymal transition in neoplastic cells [28].

Our findings regarding *TGFB1* gene expression cannot be clearly interpreted, as the expression of the *TGFB1* gene depends on many factors, including growth factors, hormones, cytokines and TGF-β itself. Cui et al. report that, during the process of carcinogenesis, the role of TGF-β may change depending on the stage of disease. It was found that, in the later stages of carcinogenesis, the cytokine is involved in tumor progression and migration of tumor cells beyond their original location [29,30]. Therefore, the differences in *TGFβ1* gene expression observed between the SW480 cell line and the SW620 line may be influenced by the level of advancement of the neoplastic changes from which both lines originate.

Nevertheless, in all assessed colorectal cancer cell lines treated with 1 µM 5-FU solution, at both exposure times, a decrease in *TGFB1* gene expression was observed. Hence, the use of low concentrations of 5-FU in the treatment of colorectal cancer in the early stages of cancer development may limit the antitumor effect of TGF-β. The remaining concentrations of 5-FU increased *TGFB1* expression in the SW480 and SW620 cells after 48 h of exposure. The exception is 100 μM 5-FU, in which no change in *TGFB1* gene expression was observed in the SW620 line. In the CACO-2 line, 5-FU increased *TGFB1* gene expression at the two highest tested concentrations. In the SW480 cell line, it increased *TGFB1* expression at the longer exposure time. Finally, in the SW620 line, an increase was only observed at 10 μM and 100 μM 5-FU.

Romano et al. reported increased *TGFB1* expression in resistant HCT116p53KO cells after 72 h of exposure to a 200 μM 5-FU solution. Cells resistant to 5-FU were found to increase SMAD3 signaling, which in turn enhances *TGFB1* expression [29]. Kensar et al. also confirm increased *TGFB1* expression in colorectal cancer cells in the early stages of carcinogenesis after exposure to 5-FU compared to controls [31].

In addition, studies suggest that the addition of TGF-β pathway or TGF-β receptor inhibitors may sensitize colorectal cancer cells to 5-FU [32]. Amerizadeh et al. reported that crocin synergistically enhanced the antiproliferative activity of 5-FU both in vitro and in vivo. Additionally, it has been shown that crocin, and especially its combination therapy with 5-FU, can dramatically reduce the number and size of tumors [33].

The present study examined the effect of 5-FU concentrations on the expression of the *TGFB1* gene. The results are promising and invite further research on the role of TGF-β1 in providing resistance to 5-fluorouracil. It is worth noting that these studies concerned the evaluation of the expression of the *TGFB1* gene in vitro; as such, further in vivo studies are needed to account for differences in the metabolic activity between cells in culture and those found in the living organism. Further research may help to develop better strategies for treating colorectal cancer, and may also help to overcome the problem of cancer cell resistance to 5-FU.

## 5. Conclusions

In colorectal cancer, SMAD4 gene expression is involved in inducing resistance to 5-FU-based therapy. The use of 5-FU at higher concentrations and for prolonged exposure times may affect SMAD4 gene expression and thus increase the effectiveness of therapy. Chemoresistance to 5-FU is a common phenomenon in colorectal cancer patients, especially in advanced stages of the disease. In addition, the use of high drug doses in these patients carries a risk of adverse effects which can be life threatening. Further clinical trials are needed to establish whether the concentrations of 5-FU found to influence cell viability in the present study cause acute toxicity.

## Figures and Tables

**Figure 1 bioengineering-10-00570-f001:**
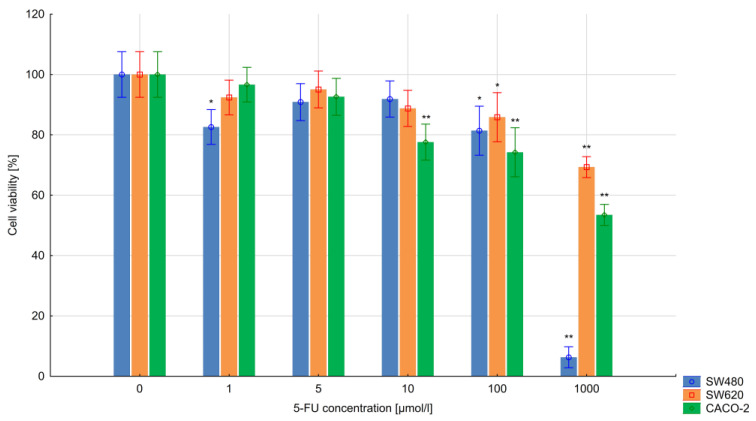
Colorectal cancer cell viability after 24 h of 5-FU exposure. Vertical bars represent 0.95 confidence intervals. Column height—mean of the group; * means a decrease in cell viability compared to the control group (ANOVA, *p* < 0.05); ** means a decrease in cell viability compared to the control group (*p* < 0.01).

**Figure 2 bioengineering-10-00570-f002:**
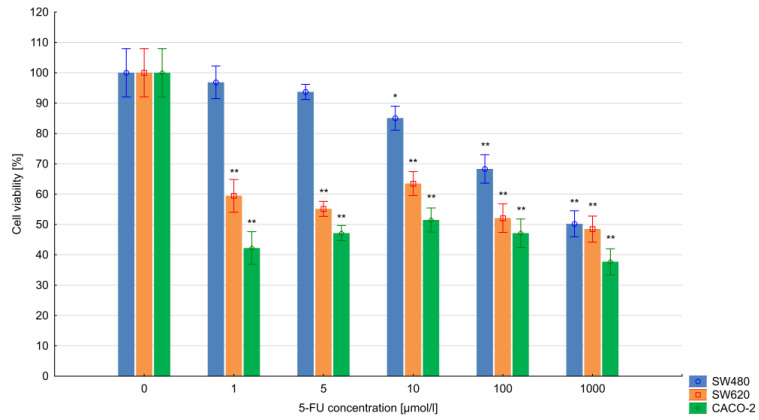
Colorectal cancer cell viability after 48 h of 5-FU exposure. Vertical bars represent 0.95 confidence intervals. Column height—mean of the group; * means a decrease in cell viability compared to the control group (ANOVA, *p* < 0.05); ** means a decrease in cell viability compared to the control group (*p* < 0.01).

**Figure 3 bioengineering-10-00570-f003:**
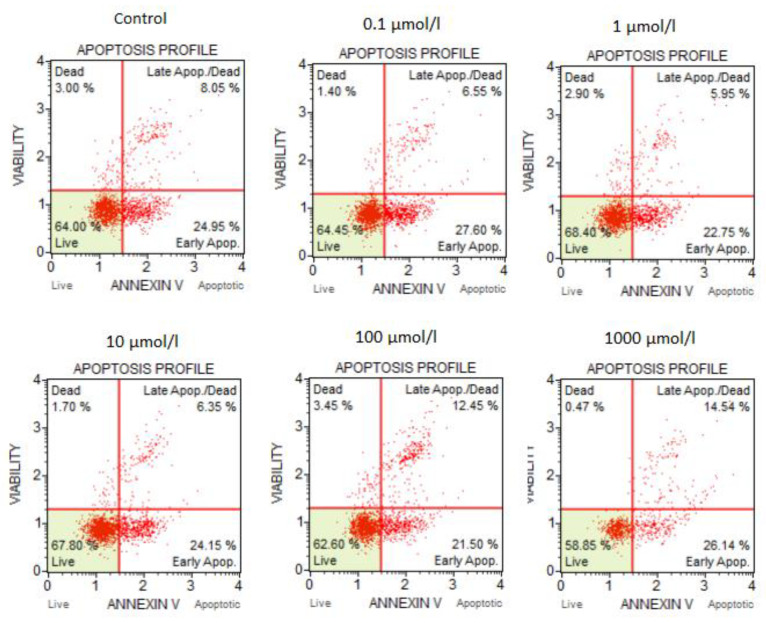
Effect of 5-FU on the induction of apoptosis.

**Figure 4 bioengineering-10-00570-f004:**
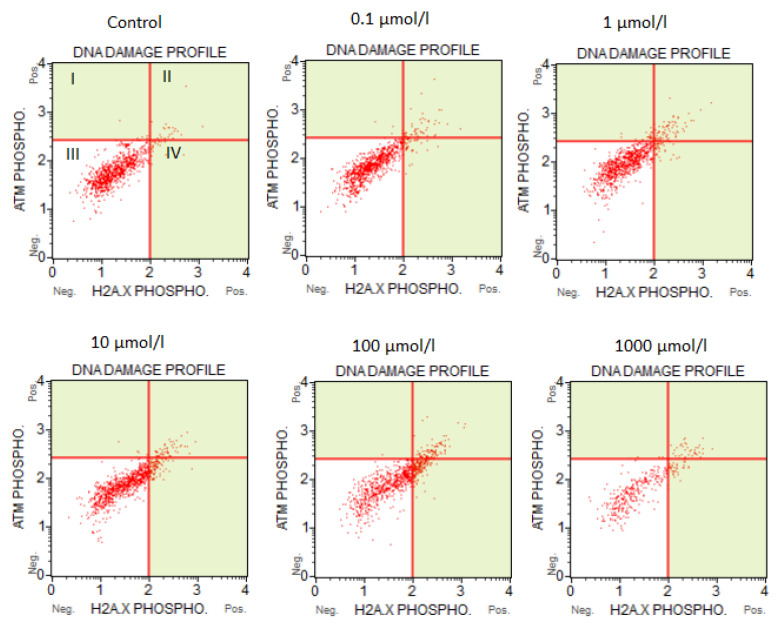
Effect of 5-FU on the induction of DNA damage. I—ATM-activated cells, II—cells with DNA double-strand breaks (dual activation of both ATM and H2A.X), III—cells without DNA damage, IV—H2A.X-activated cells.

**Figure 5 bioengineering-10-00570-f005:**
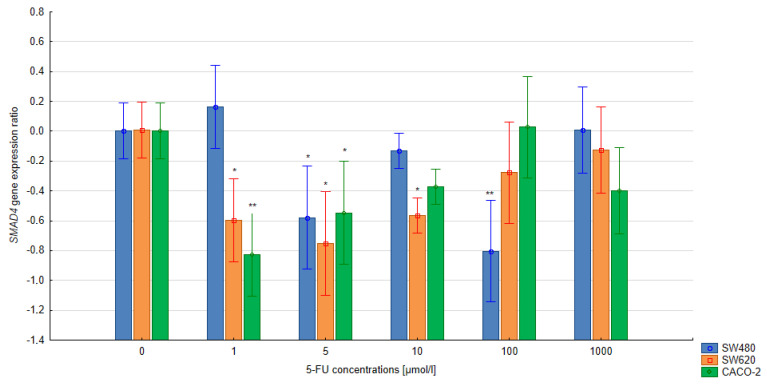
Effect of 5-FU on SMAD4 gene expression after 24 h exposure. Vertical bars represent 0.95 confidence intervals. Column height—mean of the group; * means a decrease in gene expression compared to the control group (*p* < 0.05); ** means a decrease in gene expression compared to the control group (*p* < 0.01).

**Figure 6 bioengineering-10-00570-f006:**
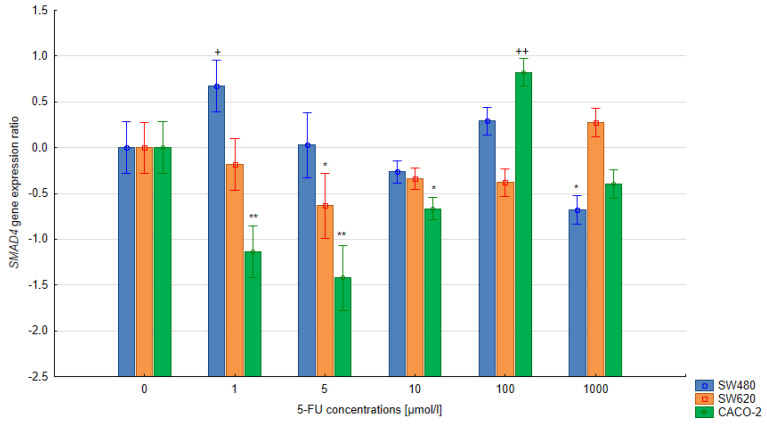
Effect of 5-FU on *SMAD4* gene expression after 48 h exposure. Vertical bars represent 0.95 confidence intervals. Column height—mean of the group; * means a decrease in gene expression compared to the control group (*p* < 0.05); ** means a decrease in gene expression compared to the control group (*p* < 0.01); + means an increase in gene expression compared to the control group (*p* < 0.05); ++ means an increase in gene expression compared to the control group (*p* < 0.01).

**Figure 7 bioengineering-10-00570-f007:**
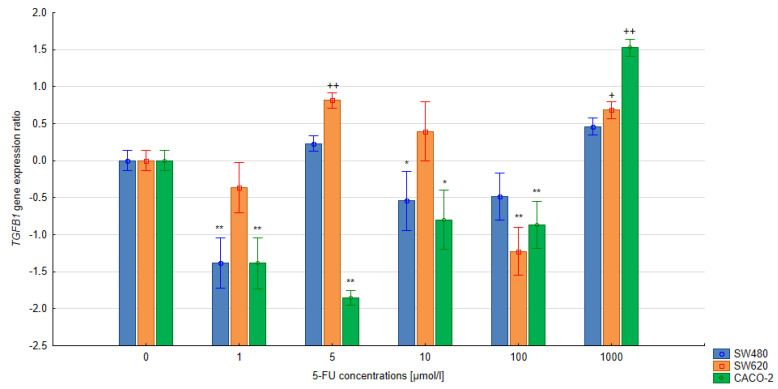
Effect of 5-FU on *TGFB1* expression after 24 h exposure. Vertical bars represent 0.95 confidence intervals. Column height—mean of the group; * means a decrease in gene expression compared to the control group (*p* < 0.05); ** means a decrease in gene expression compared to the control group (*p* < 0.01); + means an increase in gene expression compared to the control group (*p* < 0.05); ++ means an increase in gene expression compared to the control group (*p* < 0.01).

**Figure 8 bioengineering-10-00570-f008:**
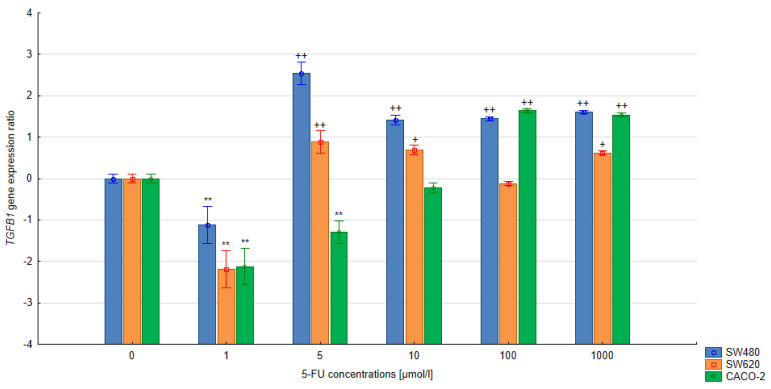
Effect of 5-FU on *TGFB1* gene expression after 48 h exposure. Vertical bars represent 0.95 confidence intervals. Column height—mean of the group; ** means a decrease in gene expression compared to the control group (*p* < 0.01); + means an increase in gene expression compared to the control group (*p* < 0.05); ++ means an increase in gene expression compared to the control group (*p* < 0.01).

**Figure 9 bioengineering-10-00570-f009:**
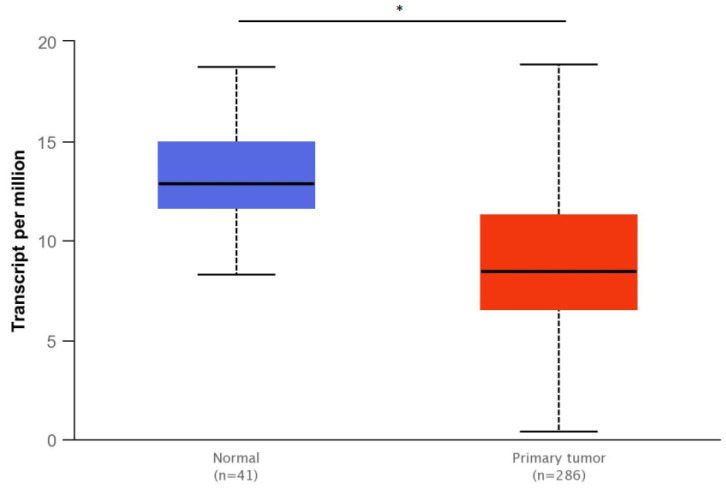
Expression of *SMAD4* in colon adenocarcinoma (COAD) compared to healthy tissue (UALCAN database). Significant decreased *SMAD4* gene expression was found in the intestinal tumor tissue in comparison with normal tissue (*p* < 0.01). TCGA samples: normal (*n* = 41), max = 18.717, upper quartile = 14.923, median = 12.887, lower quartile = 11.639, min = 8.312; primary tumor (*n* = 286), max = 18.855, upper quartile = 11.256, median = 8.45, lower quartile = 6.564, min = 0.395; * indicates a statistically significant difference between expression of *SMAD4* gene and the type of colon tissue—normal or tumor (*p* < 0.01).

**Figure 10 bioengineering-10-00570-f010:**
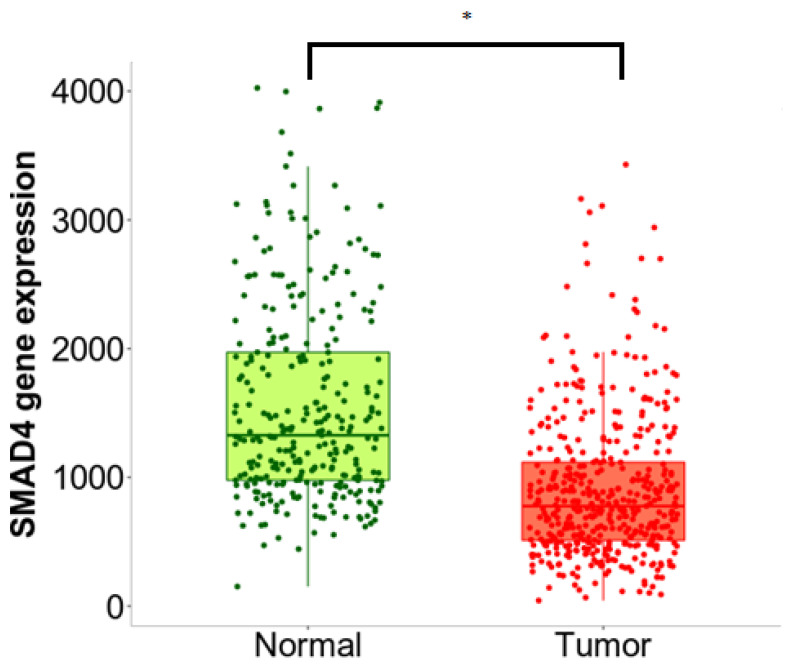
*SMAD4* gene expression analysis in colon adenocarcinoma and noncancerous tissues using RNA-Seq based data by TNMplot. A significantly lower level of *SMAD4* gene expression was observed in the tumor tissue; * indicates a statistically significant difference (*p* < 0.01).

**Figure 11 bioengineering-10-00570-f011:**
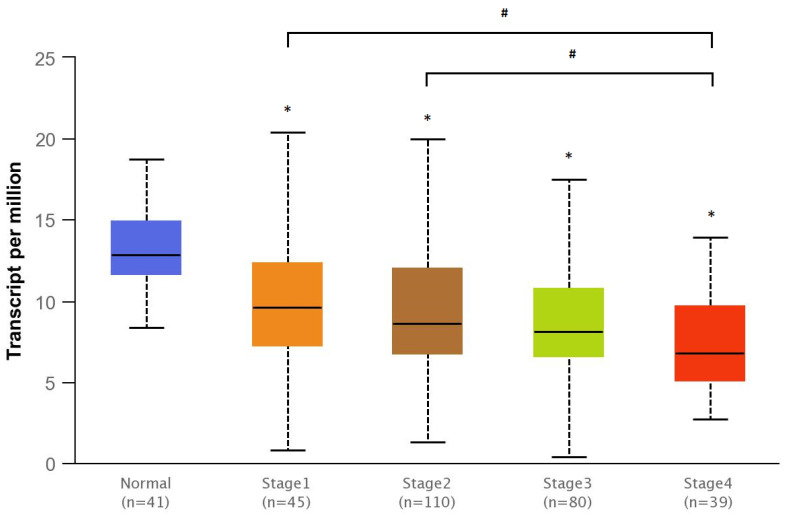
Expression of *SMAD4* in colon adenocarcinoma (COAD) according to cancer stage (UALCAN database); * indicates a statistically significant difference to the control group (*p* < 0.01); # indicates a statistically significant difference (*p* < 0.01).

## Data Availability

The data presented in this study are available on request from the corresponding author. The data are not publicly.

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
