# Peer review of "Assessment of the Influence of 5-Fluorouracil on SMAD4 and TGFB1 Gene Expression, Apoptosis Induction and DNA Damage in Human Cell Lines"

_bioengineering, 2023, doi:10.3390/bioengineering10050570_

Round 1

Reviewer 1 Report

I would like to recommend this manuscript for publication after minor revision:

1. The statistical charts in this article are mostly a mixture of bar charts and line charts, which can make reading difficult for readers. Can it be split? Authors can also consider whether it is necessary to mix bar and line charts.

2. In the article, both tables and statistical charts are mixed and grouped into one diagram. Will separate presentation affect the interpretation of the results?

3. In the article, there are also cases where Figure A is on one page and Figure B is on another page. Why not label these diagrams separately, such as Figure 1. and Figure 2 And so on.

4. In the figure legends, it should be indicated (mean±SD, N=?).

5. A recent reference about colorectal cancer is suggested for the Introduction <Colorectal Cancer and Adjacent Normal Mucosa Differ in Apoptotic and Inflammatory Protein Expression, Engineered Regeneration 2 (2021) 279-287.>

Reviewer 2 Report

The manuscript is dealing with an interesting topic which is assessment of possible influence of 5-FU on changes in the ex-22 pression of the SMAD4 and TGFB1 genes. The manuscript was well written and require certain modifications to be performed through the following comments;  

 1-   Please write the full name of SMAD4 before using this abbreviation all over the text.

2- Page 2 lines 67 and 68 the authors mentioned the following information; “SMAD4 loss is also associated with chemoresistance to 5-fluorouracil (5-FU), 67 used in the treatment of colorectal cancer”. On what scientific basics was this information built on? The reference(s) for this paragraph SHOULD be submitted.

 3-  Page 2 lines 85 and 86, the authors mentioned the following “In addition, the level of gene expression may also depend on the concentration of the drug used. A reference(s) for this information SHOULD be submitted.

 4-  Page 2 line 91 to 94 the authors wrote “Our findings confirm that SMAD4 expression is a prognostic factor in patients……………………………………” directly after the aim of the study which is not the right place to be mentioned here this SHOULD be shifted to the results section.

 5-  The other demographic data than age and race, medical, and medication history of the patients from which the tissues were sampled were missed. Please submit these information.

 6-  Page 3 line 121 in Materials and Methods section the title “Assessment of Cell Viability by MTT Assay”. Please write MTT in full term and not as abbreviation.

 7-  Study flow chart is missed. Please add this chart which summarize the study flow.

 8-  Study design is missed. Clarify?

 9-  Ethics committee approval was required as the study included biopsy from patients. Please clarify this point accurately.

 10- The references of the used concentration of 5-FU (0.1, 1,10,100 and 1000 μmol/l). SHOULD be submitted.

 11-  Page 6 lines 256 to 260 to be transferred under section recommendation and/or limitation.

 12-  The titles of the figures 1,2,3, and 4 are very summarized, a full explanation of the results in the figure are required to be added as points under each figure.

 13-  In the Discussion section, page 13 line 396 at the beginning of the paragraph write full name of “5-FU” as “5- Fluorouracil”

 14-  The authors are recommended to add this paragraph in the introduction showing the role of health care professionals on the outcome of chemotherapy treatment;

 “Most treatments for cancer despite beneficial effect result in side-effects which adversely affect patient quality of life (QOL), it was found that clinical pharmacist intervention resulted in reduction of treatment-related side-effects and the improvement of patients’ QOL.(1). Additionally, it was reported in a cross-sectional observational study performed on 500 cancer patients, to identify the incidence of prescribing medication errors (PME) involving chemotherapeutic agents, that all the cases contained at least one error and the risk factors predicting the prescribing errors were the protocol type, the tumor type, the toxicity type of the antineoplastic regimen which should be prevented for improvement of treatment plan(2).

References

 (1) doi: 10.1097/OP9.0000000000000023

 (2) DOI: 10.13040/IJPSR.0975-8232.7(8).3274-83

15-  In the Conclusion section the authors mentioned that “The use of 5-FU at higher concentrations and for prolonged exposure times may affect SMAD4 gene expression and thus increase the effectiveness of therapy.” The authors SHOULD discuss the increase in the incidence of chemotherapy toxicity by the use of higher concentration of 5FU and for long time on cancer patients which might lead to death.

Reviewer 3 Report

The authors present experiments where they expose three colorectal cell lines to different concentrations of 5-FU for either 24 or 48 h. They estimate the viability using MTT assay, apoptosis induction and DNA breaks for one (most sensitive of them) and follow the SMAD4 and TGFB1 gene transcription using RT-PCR. They also show bioinformatic analysis of SMAD4 gene expression and colorectal cancer stage using the UALCAN database, where the SMAD4 transcript levels are lower for primary colorectal adenocarcinoma tissue compared with normal, and in Stage 4 it is significantly lower than in Stage 1, 2 and in the normal tissue.

The authors find that the SMAD4 transcription level in the studied cell lines, especially CACO2, is indeed lower after treatment with 5-FU, especially after 48 h incubation, however the trend appears to be an increase with increasing 5-FU concentration between 5 and 100 micromolar. Similar trend between 1-100 micromolar is seen for TGFB-1 gene.

The findings are very interesting because they might point at a necessity of very careful dosing of 5-FU. My main remark would be that it is not very clear how many parallels were measured and how many independent experiments were performed. Also the presentation of data should be improved, results for each cell line grouped together and best two measured time points of response shown side-to-side, at the moment they are rather difficult to read. Second, the authors should avoid generalized statements and conclusions, especially in the results, discussion, and conclusion section. In the abstract, the ”Purpose” part takes more than a half of the takes, and the “Results” part is really short, although I do not think they are so straight forward to explain. I also miss a conclusion how such findings can be used to alter the dosing regime.

Please find below a list of other remarks, which I hope you will find helpful.

Line 27: The results should be summarized differently, and differentiate between the responses to different concentrations of 5-FU. Also all tested cell lines do not respond in the same way.

Line 90: „cancer cells”: cancer cell lines would be a more concise expression

Line 100: purchased at Sigma-Aldrich

Line 118: CO2, 2 in subscript

Line 120: you are repeating the information, just add “without phenol red” to the line 115)

Line 124: What do you mean with appropriately designed?

Line 125: 10e4

Line 126: the used or the conditioned medium

Line 132: The concentration of MTT of 5 mg/ml is quite high for this assay – please control the correctness.

Line 133: 1M HCl is quite high concentration for dissolving SDS-please control the correctness.

Line 137 and throughout the text: please put the gene names always in italics

Line 167: 10e4

Line 180: Can you please specify the name of the kit used?

Line 293: Results 3.1, and for all experiments: please indicate the number of parallels (replicates) in Material and Methods section.

Figure 2B: percentages should be indicated in the figure. Figures 2A and 2B. how many replicates were measured? Are the differences significant?

Figures 3 and 4: please change the color/pattern on the bars, these are really difficult to read. I propose that you group the results per cell line and show the response at different concentration grouped for each cell line (also in Figure 1). Decimal is sometimes comma and sometimes point (it should be point in all cases). What are error bars presenting (SD, SEM)?

Title to Figure 5: please define COAD abbreviation in the text

Figure 5,6, and 7: are they presenting the same parameter measured for different tissues? Could the y-axis be then labeled the same for all, and could they be parts of the same panel?

Line 489: Conclusions: again, the statements are too generalized. Please refer to specific findings.

Round 2

Reviewer 3 Report

The authors have dilligently corrected the manuscript, only they should really explicitly state which statistical test was used for which analysis (Please see the comments below). I have no further concerns.

Abstract:

Line 13: suppressor of mothers against…

Lines 28-34:

“Significant changes in the level of SMAD4 and TGFB1 genes expression were noted in the CACO-2, SW480, SW620 cells treated with 5-FU at various concentrations during 24-hour and 48-hour exposure The use of 5-FU at a concentration of 5 µmol/l resulted in a decrease in the expression of the SMAD4 gene in all cell lines, in both exposure times, while the concentration of 100 increased the expression of the SMAD4 gene in CACO-2 cells. The level of expression of the TGFB1 gene was higher for all cells treated with 5-FU at the highest concentrations, while the exposure time was extended to 48h.” please reword as:

“Significant changes in the level of SMAD4 and TGFB1 genes expression were noted in the CACO-2, SW480, SW620 cells treated with 5-FU at various concentrations during 24-hour and 48-hour exposure. The use of 5-FU at a concentration of 5 µmol/l resulted in a decrease in the expression of the SMAD4 gene in all cell lines, at both exposure times, while the concentration of 100 µmol/L increased the expression of the SMAD4 gene in CACO-2 cells. The level of expression of the TGFB1 gene was higher for all cells treated with 5-FU at the highest concentrations, when the exposure time was extended to 48h.”

For Figures 1 and 2: It should be stated which statistical test was used – I am not sure this is specified yet.

For Figures 5,6, 7 and 8: The paragraph 2.2.7 states SE was presented, and Figure legend text 0.95 CI. Please explain.

Figure 6 has a bar with 1 plus-sign, please amend the legend.

Figure 9,10, and 11: Is the statistical assessment of significance here as listed in the paragraph  2.2.6? Please amend the Figure text.

Line 312: II: cells with breaks in double-stranded DNA?

Lines 485-489: contributions of authors should continue in the line 488.

Author Response

Please see the attachment below. 
